# Epithelial Protein Lost in Neoplasm, EPLIN, the Cellular and Molecular Prospects in Cancers

**DOI:** 10.3390/biom11071038

**Published:** 2021-07-16

**Authors:** Jianyuan Zeng, Wen G. Jiang, Andrew J. Sanders

**Affiliations:** Cardiff China Medical Research Collaborative (CCMRC), Division of Cancer and Genetics (DCG), Cardiff University School of Medicine, Henry Wellcome Building, Cardiff CF14 4XN, UK; zengj8@cardiff.ac.uk

**Keywords:** EPLIN, molecular signalling, interactive partners, cancer progression

## Abstract

Epithelial Protein Lost In Neoplasm (EPLIN), also known as LIMA1 (LIM Domain And Actin Binding 1), was first discovered as a protein differentially expressed in normal and cancerous cell lines. It is now known to be key to the progression and metastasis of certain solid tumours. Despite a slow pace in understanding the biological role in cells and body systems, as well as its clinical implications in the early years since its discovery, recent years have witnessed a rapid progress in understanding the mechanisms of this protein in cells, diseases and indeed the body. EPLIN has drawn more attention over the past few years with its roles expanding from cell migration and cytoskeletal dynamics, to cell cycle, gene regulation, angiogenesis/lymphangiogenesis and lipid metabolism. This concise review summarises and discusses the recent progress in understanding EPLIN in biological processes and its implications in cancer.

## 1. EPLIN, the Structure and Cellular Functions

Epithelial Protein Lost In Neoplasm (EPLIN), also known as LIMA1 (LIM Domain And Actin Binding 1), was first reported some twenty years ago by Chang et al., as a differentially expressed protein in normal and cancerous cell lines, including keratinocyte cell lines HNOK and HOK18L [1]. It has since been implicated in the control of cancer cells and the progression of certain solid tumours. Despite a slow pace in understanding the biological role in cells and body systems and clinical implications in the early years following its discovery, more recently, rapid progress has been made in the mechanistic understanding of this protein in cells, the body and in a wider context of human tumours, as summarised in past review articles by Collins et al. and Wu et al. [2,3]. EPLIN has drawn increased attention over the past few years, with its roles expanding from those initially indicated in cell migration and cytoskeletal dynamics to cell cycle, gene regulation, angiogenesis/lymphangiogenesis and lipid metabolism. This concise review summarises and discusses the most recent progress in understanding EPLIN in biological processes and its implications in cancer.

EPLIN (Epithelial Protein Lost In Neoplasm) is present as two isoforms, a 600aa EPLIN-α and EPLIN-β. which has an additional 160aa at the amino terminus [4], and is generated from two distinct promoters [5]. A centrally located LIM Domain [4] and two actin binding sites, that flank on each side of it, allow EPLIN to gain the ability to bundle actin filaments [6] (Figure 1). EPLIN has previously been shown to inhibit branching nucleation through Actin-related protein 2/3 (Arp2/3), hence EPLIN is able to regulate actin dynamics [7]. Additionally, EPLIN was revealed to be a regulator to sustain adherens junctions (AJ), as it was directly linked to the cadherin–catenin complex via α-catenin [6]. Downregulation or phosphorylation of EPLIN by extracellular signal-regulated kinase (ERK) [8] can lead to disorganisation of the cytoskeleton, disassembly of the cadherin–catenin complex, activation of the Wnt-catenin signalling pathway and expressional alterations in a range of elements such as diminishment of E-cadherin and upregulation of ZEB1-promoting Epithelial-Mesenchymal Transition (EMT) [6,9]. This allows epithelial cells to lose cell–cell adhesion and apical basal polarity, and hence reorganise their own structure leading to mesenchymal characteristics in order to gain invasive potential. Zhitnyak et al. [10] elucidated that disorganisation of the actin cytoskeleton and E-cadherin-based AJs occur during earlier stages of Epidermal Growth Factor (EGF)-induced EMT in epithelial IAR-20 cells. The disorganisation is essential for the entire EMT process, which leads to loss of cell–cell contact. It is interesting to note that the expression of E-cadherin remains unchanged despite the loss of cell–cell adhesion. During these early events of EGF-induced EMT, disruption of colocalisation between EPLIN and linear AJ and phosphorylation of EPLIN is observed, which remains in line with earlier work by Zhang et al. [9,10,11]. These findings indicate that EPLIN contributes to the progression of EMT by at least disrupting the cell–cell adhesion complex, which further induces metastasis. Interestingly, EPLIN was found to be essential for cell division [12,13]; EPLIN was detected to locate at the cleavage furrow and to associate myosin II and Sept2 during the ingression period. Recruitment and accumulation of actin, myosin II, RhoA and Cdc42 were impacted during late-stage cytokinesis following EPLIN depletion. In addition, loss of EPLIN led to multinucleation, which results in cytokinesis failure [12]. Further study by Sundvold et al., also emphasised the role EPLIN plays in this key event of cell proliferation, by showing that EPLIN recruits Arv1 (ACAT-related protein required for viability 1), which supports the efficient progression of cell division [13]. Collectively, these studies suggest that EPLIN is needed for cytokinesis and crucial for recruitment of a number of essential elements during this process. The loss of EPLIN could lead to failure of this important event, enhancing the possibility of genetic instability and contributing to carcinogenesis.

## 2. New Members of EPLIN’s Interacting/Regulatory Network

To achieve EPLIN’s distinct functions in cellular events and metastatic progression, a number of interacting/regulatory partners are involved. As discussed above and illustrated in Figure 1, EPLIN cross-links and bundles F-actin due to its two actin binding sites and has the ability to inhibit the branching nucleation of actin filaments through Arp2/3 [7], while ERK could phosphorylate EPLIN on Ser362 and Ser604 [11]. EPLIN is also associated with the cadherin–catenin complex via α-catenin [6]. These characteristics not only allow EPLIN to regulate and maintain the cytoskeleton and AJs, and hence affect cells’ motility, but also contribute to EMT, enhancing the metastatic potential of tumour cells.

Evidence has shown that, as a tumour suppressor and due to its unique role in sustaining the epithelial cytoskeleton, EPLIN is deeply involved in multiple cellular processes associated with carcinogenesis and metastasis. To understand more about its role in these processes, and others, increasing research has focused on elucidating interacting or regulatory proteins of EPLIN, either upstream regulators or downstream participants. 

### 2.1. p53, a Direct Regulator of EPLIN

p53 is a universally known tumour suppressor in multiple human tumours, with mutation or reduction of p53-promoting cancer progression [14]. Since EPLIN is also characterised as a putative tumour suppressor, their relationship with each other seems fascinating and worthy of investigation. DNp73, a mutant isoform of the p53 family capable of inhibiting the expression of p73, was found to be involved in the regulation of EPLIN [15]. Steder et al. reported that DNp73 could induce downregulation of EPLIN, and this could lead to disruption of AJs and allow insulin-like growth factor 1 receptor (IGF1R) to bind its ligands. This would cause further phosphorylation of AKT and Signal Transducer and Activator of Transcription-3 (STAT-3) to downregulate E-cadherin and upregulate Slug. Morphological changes in melanoma cells are also observed upon regulation of EPLIN by DNp73, which indicates progression of EMT and enhances metastatic potential [15]. Due to the relationship between DNp73 and p53, it naturally draws attention to whether p53 also takes part in the regulation of EPLIN. Indeed, later study by Ohashi et al. [16] uncovered the regulatory association between these two tumour suppressors. The authors identified that p53, p63γ and p73β overexpression, in H1299 lung cancer and Saos-2 osteosarcoma cells, could enhance EPLIN mRNA expression. Furthermore, utilising a combination of chromatin immunoprecipitation-sequencing (ChIP-seq [17]), ChIP-PCR and reporter assays, the authors identified two p53 consensus motifs within EPLIN which facilitated transactivation of EPLIN by p53 family members. Similarly, expression of p53, p63γ and p73β was found to enhance EPLIN protein levels in H1299 cells. Interestingly, nutlin-3a treatment of wild-type, but not p53-mutated cells, also enhanced EPLIN protein expression without impacting on p63 or p73 expression in MCF7 breast cancer, LoVo colon cancer and A549 lung cancer cells, implicating p53 in inducing EPLIN protein expression. The authors went on to show a downregulation of EPLIN in breast and colorectal cancers that had p53 mutations (TCGA database). Lower levels of EPLIN were associated with significantly shorter survival periods in colorectal cancer, breast cancer and lung cancer. The authors also demonstrated, utilising a combination of nutlin-3a, EPLIN knockdown and p53 expression systems in A549 and Lu99 lung cancer cells, that p53 could suppress cellular invasion and that this suppression was partially inhibited by knockdown of EPLIN. Interestingly, in vivo administration of nutlin-3a, via the intra-peritoneal route, was able to reduce the size of established tumours (derived from subcutaneously inoculated A549 cells) but had a lesser impact on tumours from the EPLIN knockdown A549 cells. [16]. Hence, EPLIN was reported to be a target of p53, with this relationship influencing metastatic progression. These findings support EPLIN’s role as a tumour suppressor and clinical prognosis indicator.

### 2.2. hCDC14A Dephosphorylates EPLIN

EPLIN’s ability to sustain the cytoskeleton and regulate actin dynamics is attributed to its capacity to directly bundle F-actin on two actin binding sites [7] and its link to the cadherin–catenin complex to support AJs [6]. The ability of mouse EPLIN to bind and regulate actin dynamics is reported to be weakened following ERK phosphorylation of serine residues within EPLIN, induced by platelet-derived growth factor (PDGF) [8]. A later study subsequently demonstrated the capacity of EGF-induced ERK to phosphorylate human EPLIN in prostate cancer cells on Ser362 and Ser604, which pair to Ser360 and Ser602 in mouse EPLIN, leading to EPLIN ubiquitination and degradation and downregulation of E-cadherin, an essential marker of EMT progression [11]. These findings indicate a link between EPLIN, cell migration and cancer metastasis.

Chen et al. [18] conducted phospho-proteome and Biotin identification (BioID) analyses to identify that EPLIN is involved in the interacting network of human Cell Division Cycle 14A (hCDC14A). Furthermore, hCDC14A was revealed to be able to dephosphorylate EPLIN on Ser362 and Ser604 to counteract the phosphorylation induced by ERK, and influence F-actin stability via this particular feature in Hela cells [18]. Knocking down EPLIN or reducing the activity of hCDC14A, through the generation of a phosphatase dead version of hCDC14A, in the HCT-116 CRC cell line significantly decreased E-cadherin to allow the cells to acquire mesenchymal characteristics [18]. The study also demonstrated that downregulation of EPLIN and hCDC14A was associated with poor prognosis in colorectal cancer, by exploring online databases. Thus, it would appear that hCDC14A acts as a vital upstream player of EPLIN during cancerous development.

### 2.3. Paxillin–Plectin–EPLIN Complex Promotes Apical Extrusion

EPLIN has been indicated in the process of cytokinesis, in which loss of EPLIN and its interacting partners could lead to multinucleation which, in turn, may contribute to carcinogenesis [12,13]. During the early carcinogenesis period in epithelial tissue, mutations of oncogenes are crucial contributors. Kajita et al. revealed, by using mammalian epithelial cells, that when Ras-transformed cells are surrounded by normal cells, the latter will fight to lift transformed cells from the monolayer in order to eliminate them. This competitive self-defence process was described as epithelial defence against cancer (EDAC) [19]. EPLIN has also been identified as playing a role in this process, contributing to apical extrusion of RasV12-transformed Madin–Darby Canine Kidney (MDCK) cells, where it has been linked to Caveolin-1 (Cav-1) [20]. In the study, Ohoka et al. utilised Cav-1 immunoprecipitation under a number of cell culture conditions, namely, normal cell culture, RasV12-transformed cell culture and combinations of the two cultures. The study has identified EPLIN as a binding partner predominantly in the mixed culture setting and at apical and lateral membrane domains, where it was observed to be partially co-localised with Cav-1 using immunofluorescence. EPLIN was also seen in the cytosolic and RasV12 intracellular regions where Cav-1 was absent. Knocking down EPLIN by short hairpin RNA in transformed cells suppressed accumulation of Cav-1 and activation of myosin II and protein kinase A (PKA). It also impacted on the apical extrusion process. Interestingly, Cav-1 knockdown did not impact EPLIN accumulation or myosin II and PKA activity. It is interesting to note that addition of the MAPK/ERK Kinase (MEK) inhibitor U10126, or the actin polymerisation inhibitor cytochalasin D, could impact the accumulation, localisation or enrichment of Cav-1 and EPLIN. The authors also demonstrated that the accumulation of filamin in normal cells surrounding transformed cells is repressed following EPLIN or Cav-1 knockdown in RasV12-transformed cells and, similarly, EPLIN and Cav-1 enrichment in transformed cells is suppressed following filamin A knockdown in surrounding normal cells [20]. Hence, EPLIN has been implicated in this interesting, competitive process along with a number of elements, with Mitogen Activated Protein Kinase (MAPK) pathways and actin dynamics also reported to be involved. Subsequently, the mechanism behind apical extrusion was explored. Ras-Associated Protein 5 (Rab5) regulates endocytosis, which has been shown to be involved in cell migration and oncogenesis [21]. Rab5 induces endocytosis of E-cadherin and mediates EPLIN to disconnect the E-cadherin complex, potentially allowing EPLIN to subsequently interact with players such as myosin II and PKA, to promote the EDAC elimination process, in RasV12-transformed cells surrounded by normal cells [22]. It was further revealed that plectin and paxillin immunoprecipitated and partly co-localised with EPLIN in Ras-transformed cells surrounded by normal cells. Following knockdown of either of the molecules, using shRNA, apical extrusion activity was repressed and accumulation of either of the proteins in this situation was also depleted significantly [23,24]. Collectively, this would suggest that these molecules form a plectin–paxillin–EPLIN complex to regulate apical extrusion. Digging deeper into the interaction of this complex, α-tubulin is accumulated and regulated by the complex, and acetylated tubulin is also upregulated in transformed cells, when they are surrounded by normal cells. This enhanced acetylation was found to be repressed by deacetylation caused by Histone Deacetylase 6 (HDAC6), which could be regulated by paxillin [23,24]. Hence, the picture of the mechanism behind this competitive process and EPLIN has become clearer. EPLIN and its interacting molecules form a complex to support promotion of acetylation of tubulin, by mediating HDAC6 to regulate microtubule filaments and cell–cell adhesion. This would enhance apical elimination. Other molecules, such as PKA and myosin II, the MAPK pathway and the actin cytoskeleton, also help regulate the elimination process. 

### 2.4. EPLIN Regulates Cellular Functions Partly Through the FAK/Src Signalling Pathway

A recent study by our laboratories [25] demonstrated a possible interaction between paxillin, focal adhesion kinase (FAK) and proto-oncogene tyrosine protein kinase (Src). Both kinases have been demonstrated to affect focal adhesion dynamics, cell migration and cancer development via certain signalling pathways [26,27]. Collins et al. reported that EPLIN expression is reduced in prostate cancer tissue when compared to normal paired tissue from tissue microarrays. Overexpression of EPLIN-α in the PC-3 prostate cancer cell line induced significant repression of cellular growth, invasion and migration. Knocking down EPLIN in the CA-HPV-10 prostate cancer cell line, on the other hand, promoted invasion and migration [25]. Such observations are in line with the findings from an earlier study by Zhang et al. [9], supporting an argument that EPLIN influences cell function by coordinating with FAK and Src. Direct evidence for this connection subsequently came from protein microarray and Western blotting analysis, which showed significant increases in expression of p-FAK Y925, p-Paxillin Y31, total Paxillin and p-Paxillin Y118, following overexpression of EPLIN-α in the PC-3 cell line. EPLIN-α overexpressed LNCaP cell lines, on the other hand, showed that p-FAK Y397 was upregulated and p-Paxillin Y118 downregulated. Interestingly, in the PC-3 EPLIN-α overexpression cell model, an activation site, p-Src Y419, was seen to be depleted and a regulatory site, p-Src Y530, was increased. Knockdown of EPLIN in CA-HPV-10 cells using shRNA resulted in the expression of total FAK being upregulated, while Src Y419 was depleted significantly [25]. Among these activation sites, FAK Y925/Y118, Paxillin Y118/Y31 and Src Y419 have been identified to be key phosphorylation sites and have significant impacts on cellular functions and cancer development [25,26,28].

Furthermore, the study by Collins et al. explored the impact of the EPLIN, Src, FAK relationship on cellular invasion and migration. Invasive capability in PC-3 control groups was repressed when FAK and Src were inhibited, but no significant differences were detected in EPLIN-α-overexpressed PC-3 cells lines. Migration was reduced at some time periods following inhibition of FAK or Src in both the control and EPLIN-α-manipulated groups in PC-3 cells, though the impact appeared generally reduced in the EPLIN-α overexpression groups. In LNCaP models, migration was suppressed in EPLIN-α-overexpressed cell lines when FAK was inhibited, whereas significant changes within the control group following treatment were not observed, while the migration of both the control and EPLIN-α-overexpressed cell lines were affected following Src inhibition. The invasive capacity of LNCaP control cells was significantly reduced following Src inhibition, though this trend was not found to be significant in the EPLIN-α overexpression LNCaP line. On the contrary, invasion and migration were promoted significantly while knocking down EPLIN in CA-HPV-10 cells compared to its control group. When Src was inhibited, reduction of migrated and invasive cells was observed in both control and EPLIN-knockdown cell lines, while invasion and migration abilities were only significantly reduced in EPLIN-knockdown CA-HPV-10 cells following FAK inhibition [25]. Taken together, EPLIN expression is downregulated in prostate cancer when compared to normal tissues. Its impact on cellular functions in prostate cancer cells may be achieved through regulation of FAK/Src signalling pathways, adding insights to the potential mechanism behind this tumour suppressor.

Recent scientific focus on the tumour suppressor EPLIN has aided the understanding of this important molecule and its wider role and interactions in cancerous epithelial cells. Such hypothetical signalling pathways are summarised in Figure 2. 

### 2.5. MicroRNAs (miRs) as Regulators of EPLIN

Liang et al. reported a microRNA, miR-93-5p, as a novel upstream regulator of EPLIN [29]. Inhibition of miR-93-5p significantly upregulates expression of EPLIN, whilst enhancing the expression of miR-93-5p leads to downregulation of EPLIN in human umbilical vascular endothelial cells (HUVECs), indicating a negative correlation between the two elements. Similarly, luciferase reporters confirmed that miR-93-5p manipulates expression of EPLIN by binding to its 3′-UTR sequence. This relationship was further revealed to be associated with migration and angiogenesis in HUVECs (discussed later) [29]. More recently, work by Dart et al. has identified miR-221 as a potential regulator of EPLIN. In their study, the authors generated miR-221-deleted PC-3 cells (PC3 miR-221 del) and observed reductions in aggressive traits such as cellular migration and invasion in these cells. Furthermore, characterisation of PC3 miR-221 del cells, for proteins associated with processes such as motility, invasion and EMT, identified changes in EPLIN expression, with enhanced expression of the alpha isoform noted. Taken together, this study supports the association of EPLIN with these key processes and adds information regarding the complex regulatory networks related to EPLIN [30].

### 2.6. EPLIN Interaction with LUZP1 and NPC1L1 and Implications in other Biological Processes and Disease

From earlier studies, one of the most significant findings concerning EPLIN is that it regulates actin dynamics by colocalising with actin filaments and other actin structure regulators and cross-linking actin filaments, inhibiting branched nucleation through Arp2/3, further to affecting cells’ motility and migration in order to promote cancer development [4,7]. Except for its notable impact on solid tumours, EPLIN was discovered to be involved in cellular activities in noncancerous tissues. Tsurumi et al. reported strong expression of EPLIN in mesangial cells and demonstrated EPLIN downregulation in mesangial proliferative nephritis in vivo [31]. EPLIN colocalises at focal adhesions with paxillin and their interaction takes part in stabilising focal adhesion in mesangial cells. PDGF-induced MEK/ERK signalling is responsible for disruption of the EPLIN–paxillin complex and translocation of EPLIN from focal adhesion sites to peripheral ruffles. In addition, depletion of EPLIN results in disorganisation of focal adhesion and enhancement of cells’ migration via PDGF [31].

A recent study by Goncalves et al. revealed that EPLIN is also involved in another cellular activity, ciliation, by interacting with LUZP1 (Leucine Zipper Protein 1) [32]. Cilia are membranous protrusions which originate from centrosomes via complicated mechanisms including the cytoskeleton, membrane traffic, etc. Cilia take part in certain sensory and motional biological functions whose dysregulation could lead to ciliopathies, including blindness, cystic kidneys, etc. [32,33,34]. LUZP1 has been reported as a negative regulator of ciliogenesis and a positive regulator of actin dynamics [33,35]. 

Goncalves et al. identified EPLIN as a potential interacting partner of LUZP1 by conducting BioID assays in cycling and serum-deprived HEK293 cells [32]. The relationship was confirmed through conducting co-IP assays using GFP/FLAG vectors in RPE-1 cells or HEK293 cells. Here, the authors revealed that LUZP1 and EPLIN interact with each other via the C-terminal of LUZP1, as GFP-tagged LUZP1 pulled down both isoforms of EPLIN in RPE-1 cells, GFP-EPLIN isoforms/FLAG-EPLIN-β pulled down LUZP1 in RPE-1 cells and HEK293 cells, respectively. GFP-EPLIN-β was able to pull down FLAG-tagged full length and C-terminal LUZP1 in HEK293 cells. FLAG-EPLIN-β and FLAG-LUZP1 are able to pull down actin. Intriguingly, immunofluorescence revealed that both EPLIN and LUZP1 co-locate with actin filaments in RPE-1 cells. However, LUZP1 locates at centrosome and basal regions, EPLIN-α locates mainly at the leading edge where membrane ruffles occur, while EPLIN-β mainly locates along with actin filaments, indicating a possible functional correlation between these proteins. Furthermore, accumulation of ciliated cells and longer primary cilia were observed, following siRNA-mediated knockdown of EPLIN/LUZP1 in RPE-1 cells, along with increased expression of myosin Va after immunofluorescence analysis. Aberrant ciliation, caused by cytochalasin D, could be counteracted by overexpressing EPLIN and LUZP1, while accumulation of Arp2 was also observed. Hence, LUZP1 interaction with EPLIN contributed to ciliation regulation, potentially partly through regulating actin structure [32].

Given that EPLIN participates in the regulation of ciliation progression, interacting with LUZP1 [32], and that depletion and dysfunction of cilia can result in diseases such as blindness, cystic kidneys, etc. [32,34], this demonstrates EPLIN’s implication and importance in other biological processes, not only in carcinogenesis and tumour development. Indeed, another study by Zhang et al. [36] reported that EPLIN is associated with cholesterol absorption in intestines. A Chinese Kazakh family, with inherited low levels of low-density lipoprotein cholesterol (LDL-C) in plasma, has been established by the authors as a study model. A mutation of EPLIN, LIMA1-K306fs, which includes a frameshift variant on exon-7, was identified to be a potential candidate associated with LDL-C in the family, by using whole-exome sequencing and sanger sequencing. The authors identified that individuals who express LIMA1-K306fs have a significantly lower level of LDL-C and campesterol:lathosterol ratio when compared to those who do not. Through analysis of a larger cohort, the study identified that another mutation of EPLIN, LIMA1-L25I, also has a similar effect, although the impact on LDL-C levels in these carriers was not as great as in the K306fs groups. Moreover, the team developed a mouse experimental model and discovered that silencing EPLIN in the intestines of the mice led to the downregulation of cholesterol uptake, plasma cholesterol, liver ^3^H-cholesterol and plasma ^3^H-cholesterol, when compared to control groups. Hence, these implicate EPLIN as a potential positive regulator in LDL-C levels and intestinal cholesterol absorption in humans and mice [36]. To investigate the possible mechanism behind this interesting function, EPLIN was found to bind and colocalise with myosin Vb and Niemann-Pick C1-Like 1 (NPC1L1), which are known to be essential to cholesterol absorption [36,37] on the brush border in mice intestines [36]. Furthermore, knocking down EPLIN in CRL1601 cells led to weakened association between myosin Vb and NPC1L1. EPLIN is seen to interact with both proteins on certain regions, namely the Q_1277_KR residues of NPC1L1 and the C_164_LG residues of EPLIN being responsible for their interaction, while the 21 to 40 amino acid regions of myosin Vb and the 491 to 511 amino acid regions of EPLIN contributed to the interaction of these two molecules. Hence, the authors of the study propose that EPLIN might function as a connecting bridge between myosin Vb and NPC1L1. Depletion of EPLIN or myosin Vb, as well as mutation of EPLIN using CRISPR-Cas9 in CRL1601 cells, led to disruption of NPC1L1 translocation from the endocytic recycling compartment to the plasma membrane. Furthermore, the authors identified that the NPC1L1–EPLIN complex is needed for cholesterol absorption, as disrupting the complex led to a weakened rate of transportation of NPC1L1 in vitro and in vivo and attenuated liver cholesterol and plasma total cholesterol levels in vivo [36]. Therefore, EPLIN was found to be associated with LDL-C plasma levels, whose high concentration represents a significant risk factor for cardiovascular disease, and to play a role in cholesterol absorption by interacting with NPC1L1 and myosin Vb.

Taken together, the above studies have increased our understanding of the interacting and regulatory networks associated with EPLIN, which are outlined in Table 1. This has added to previously established networks from earlier studies which have been summarised and discussed in past reviews by Collins et al. [2] and Wu et al. [3]. Thus, EPLIN has been linked with a wide range of partners to achieve multiple functions, not only in carcinogenesis and tumour development, but also in maintenance of focal adhesion in mesangial cells, cilia formation and cholesterol absorption. Interestingly, depleted or mutant EPLIN impacts the interactions with NPC1L1 to weaken cholesterol absorption in intestines, potentially decreasing the risk of high LDL-C-related diseases, which provides a different picture of this tumour suppressor, as its downregulation in cancer cells often leads to promotion of cancer developments. 

## 3. Role of EPLIN in Endothelial Cells, Angiogenesis and Lymphangiogenesis

EPLIN was first observed when looking at differential expression between cancer and normal cells [1,4]. It had been largely investigated regarding its cellular function and interacting/regulatory partners, mainly in epithelial-derived cancer cells, leading to its labelling as a tumour suppressor. Angiogenesis is essential in cancer development and progression, for its role in blood, nutrient and oxygen supply and for acting as an escape route for disseminating cancer cells [38]. However, EPLIN’s role in angiogenesis was not in the spotlight in early studies. EPLIN bundles actin filaments and connects to the cadherin–catenin complex via α-catenin. It inhibits branch nucleation through Arp2/3 contributing to sustaining the cytoskeleton and cell–cell adhesion in epithelial cells [6,7]. Similar cell junction activities in endothelial cells, which regulate endothelium integrity, are crucial for angiogenesis [39,40]. Through overexpression of EPLIN-α in human endothelial HECV cells and conducting wound-healing and Matrigel adhesion assays, Sanders et al. reported that migration and adhesive ability were significantly downregulated when compared to control groups. Furthermore, the impact of EPLIN-α overexpression in HECV cells was also tested, using tubule formation assays and through the co-injection of either overexpressed EPLIN-α or plasmid control HECV cells with MDA-MB-231 cells in mice, demonstrating a role in inhibiting tubulelike structure formation in in vitro and in vivo tumour development [41]. Hence, EPLIN has the potential to affect angiogenesis, with a number of studies focused on EPLIN’s role and mechanism in endothelial cells and angiogenesis.

An early study by Chevin-Petinot et al. [39], utilising a range of techniques, including confocal microscopy and immunoprecipitation assays in HUVEC cells, demonstrated that EPLIN co-locates, or is associated with, actin filaments, VE-cadherin, α/β -catenin and vinculin at cell junction regions. Knocking down EPLIN leads to location changes of vinculin, highlighted by the delocalisation of vinculin from cell–cell junctions in EPLIN suppressed cells. A GST pull-down assay showed EPLIN links to the VE–cadherin–catenin complex via α-catenin, although suppressing EPLIN did not affect adhesion, migration and proliferation of HUVECs. The authors also conducted tubule formation assays to show that suppression of EPLIN negatively impacted capillary network formation, enhancing breakage events in comparison to controls [39]. Another study reported that miR-93-5p, a microRNA which promotes migration, proliferation and angiogenesis in HUVECs, acts upstream of EPLIN, demonstrating regulation of EPLIN by miR-93-5p, and that siRNA suppression of EPLIN was able to negate the impact of miR-93-5p antisense oligos on HUVEC migration and lumen formation [29]. This may imply that the role of EPLIN in HUVECs cellular functions could be achieved through interaction with, or regulation by, other elements. Taken together with other studies, such data suggest a complex and key role for EPLIN in regulating angiogenesis. 

EPLIN has two isoforms which generate from two distinct promoters [5], only differing in the N-terminal region, in which EPLIN-β has an additional 160aa [4], and share a central-located LIM domain and two actin binding sites [4,7], which are essential for EPLIN’s function. Expression of EPLIN-α is frequently diminished in cancer, while EPLIN-β has been reported to remain the same or slightly increase [5]. A recent study identified that the two isoforms play different roles in endothelial cell dynamics [42]. In their study, Taha et al. focused on the specific function of individual isoforms in endothelial cells. The authors reported expression of both α and β isoforms in pig aorta and cava vein endothelial cells, noting enhanced expression of EPLIN-β in aorta compared to cava vein samples, with little change noted in EPLIN-α between samples. Interestingly, they reported enhanced EPLIN-β expression in confluent HUVECs following sheer stress application, whilst EPLIN-α levels were not significantly impacted. Additionally, EPLIN-α but not β isoform expression was noted to correlate with HUVEC confluence, with its expression, protrusion formation and migration velocity reducing in confluent compared to growing cultures. EPLIN expression was observed to be localised to cell junctions and stress fibres, with analysis of tagged isoform localisation also indicating their presence at cell junctions and stress fibres. Importantly, the authors observed an isoform-specific role for EPLIN-α, noting its presence within or in close proximity to branched actin filament networks at membrane protrusions, such as classical lamellipodia (cLP) and junction-associated intermittent lamellipodia (JAIL). However, EPLIN-β was mostly negative at protrusions and was noted to be potentially involved in retraction. EPLIN-α was also observed to be more dynamic than EPLIN-β, collectively supporting differential isoform roles in endothelial cells and the proposal that EPLIN-α contributes to cell migration and junction remodelling, whereas EPLIN-β contributes to filament stabilisation [42].

Subsequently, Taha et al. investigated the role of EPLIN-α at cLP and JAIL and its relationship with the Arp2/3 complex [42]. JAIL is responsible for forming and developing endothelial cells’ junction sites, where vascular endothelial cadherin (VE-cadherin) localises. This dynamic regulation by JAIL contributes to migration and junction dynamics related to angiogenesis [43]. Spinning disc confocal microscopy (SpDM) demonstrated that the distance between EPLIN-α and the Arp2/3 complex narrowed during protrusion extension, and that the eventual overlap of these proteins resulted in the halting of protrusion movement [42]. This coincided with EPLIN-α and Arp2/3 disconnection with, and loss of, actin filaments. Taken together with the findings that EPLIN isoforms associated with the Arp2/3 complex during pull-down assays implicated a potential role for EPLIN in protrusion termination via the Arp2/3 complex interaction. In support of this, the authors demonstrated that CK666-mediated inhibition of Arp2/3 resulted in an inhibition of protrusion formation and EPLIN-α’s materialisation at the membrane. In keeping with this, targeting EPLIN using siRNA resulted in both increased protrusion size and duration and an enhancement of migration velocity in normal medium. Furthermore, overexpression of EPLIN-α-EGFP resulted in EPLIN-α-EGFP appearance at, and disruption of, JAIL-like structures, preventing protrusion expansion, and also enhanced the formation of filopodia, indicating enhancement of actin dynamics. The authors suggested that enhanced EPLIN-α results in elevated binding to the Arp2/3 complex and JAIL formation termination, potentially impacting VE-cadherin dynamics. Consistent with this, they demonstrated altered VE-cadherin dynamics in EPLIN-α overexpression HUVEC cells, observing intracellular gaps and disrupted VE-cadherin at cell contacts, together with decreased migration and barrier function [42]. Interestingly, the authors demonstrated a prominent role of EPLIN-β in stress fibre induction and stabilisation, demonstrating that stress fibre formation is enhanced substantially following EPLIN-β overexpression (and to a lesser extent EPLIN-α overexpression) in endothelial cells. Furthermore, EPLIN-β was found to protect stress fibres from depolymerisation resulting from Y27632 ROCK inhibitor treatment, with minimal disassembly noted in EPLIN-β overexpression cells compared to control, or EPLIN-α overexpression cells, following such treatment [42]. This important study has shed light on the isoform specific role of EPLIN in endothelial cells, highlighting EPLIN-α’s role in regulating protrusion progressions by interacting with Arp2/3 and regulating JAIL formation, VE-cadherin dynamics and implication in migration and barrier function, whereas EPLIN-β plays key roles in induction and stabilisation of stress fibres. Currently, our understanding of the isoform specific role of EPLIN is limited. Further investigations in this vital area are needed to enhance our understanding of this important molecule across many biological areas including cancer development and progression.

Taken together, EPLIN takes part in the regulation of endothelial dynamics by binding to VE-Cadherin via α-catenin and actin filaments [39]. Depletion of EPLIN could be induced by miR-93-5p, which contributes to elevation of cellular migration and lumen formation [29], whereas overexpression of EPLIN-α has been shown to reduce cell migration and tubule formation in HECV endothelial cells [41]. Hence, EPLIN appears to play key roles in endothelial cells and associated processes. A summary outlining the potential role of EPLIN in endothelial cells is outlined in Figure 3.

## 4. The Clinical Aspects of EPLIN in Human Cancers

EPLIN has been implicated in playing a role in the development and progression of solid cancers. Keeping in line with the findings that EPLIN contributes to maintaining the epithelial cytoskeleton, regulating metastatic progression and cell cytokinesis, there has been strong scientific focus on EPLIN and its implications in multiple types of cancer including oral cancer [1,4], breast cancer [4,44], prostate cancer [2,9], squamous cell carcinoma of head and neck (SCCHN) [9], lung cancer [45], oesophageal cancer [46], ovarian cancer [47], colorectal cancer (CRC) [9,16,18,48,49] and most recently gastric cancer [50]. EPLIN has also been revealed to be associated with multiple cellular functions such as cell motility, migration, invasion and proliferation, in a number of different cancer cell lines [2,9,25,44,45,46,47,51]. Our lab, as a contributor to EPLIN research for over a decade, has demonstrated downregulation of EPLIN in multiple tumour tissues, its association with cellular function and clinical significance [2,44,45,46,47,51]. Furthermore, important work by Zhang et al. revealed that a lower level of EPLIN is related to poor prognosis and is implicated in chemo drug resistance in prostate cancer [9]. The initial finding that EPLIN may have a role to play in drug resistance has recently been supported by a more clinically oriented finding in gastric cancer [50], in that higher expression was generally seen in cancers that responded to neoadjuvant chemotherapy (NAC) compared to those that did not, though this was not significant. Furthermore, a greater overall survival distribution in patients displaying higher expression of EPLIN was observed, based on responsiveness to NAC. Those patients with high levels of EPLIN, who responded to neoadjuvant therapies, had a marked longer survival. Together, such work highlights the significant role played by EPLIN in multiple aspects associated with cancer progression, further supporting its identified role as a tumour and metastasis suppressor.

## 5. Conclusions

Focus on EPLIN, since its discovery several decades ago, has implicated it as a key regulator of several important cellular processes and as a tumour suppressor. Furthermore, research has identified a number of interacting partners or regulatory mechanisms associated with EPLIN. In this review, we provide an update to previous review articles [2,3] focused on this important molecule, summarising major findings related to EPLIN in recent years. Such works have revealed that EPLIN contributes to not only metastasis-relevant processes such as EMT with other novel interacting/regulatory proteins, but also apical elimination, ciliation, cholesterol absorption, angiogenesis and endothelial cell dynamics, providing additional routes to explore EPLIN’s implication and mechanisms in a wider range of fields. Research into the field of EPLIN is intensifying to explore the full implications of EPLIN in multiple arenas. Widening our understanding of this important molecule will enhance its therapeutic potential and aid in the design of novel therapeutic strategies.

## Figures and Tables

**Figure 1 biomolecules-11-01038-f001:**
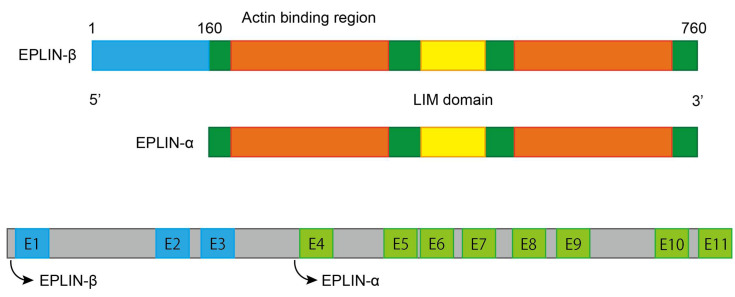
Schematic structure of EPLIN. Structural information is adapted from [5,6] and figure is designed and created by using Adobe Illustrator 2021Version 25.2.1 (Adobe Inc., San Jose, CA, USA).

**Figure 2 biomolecules-11-01038-f002:**
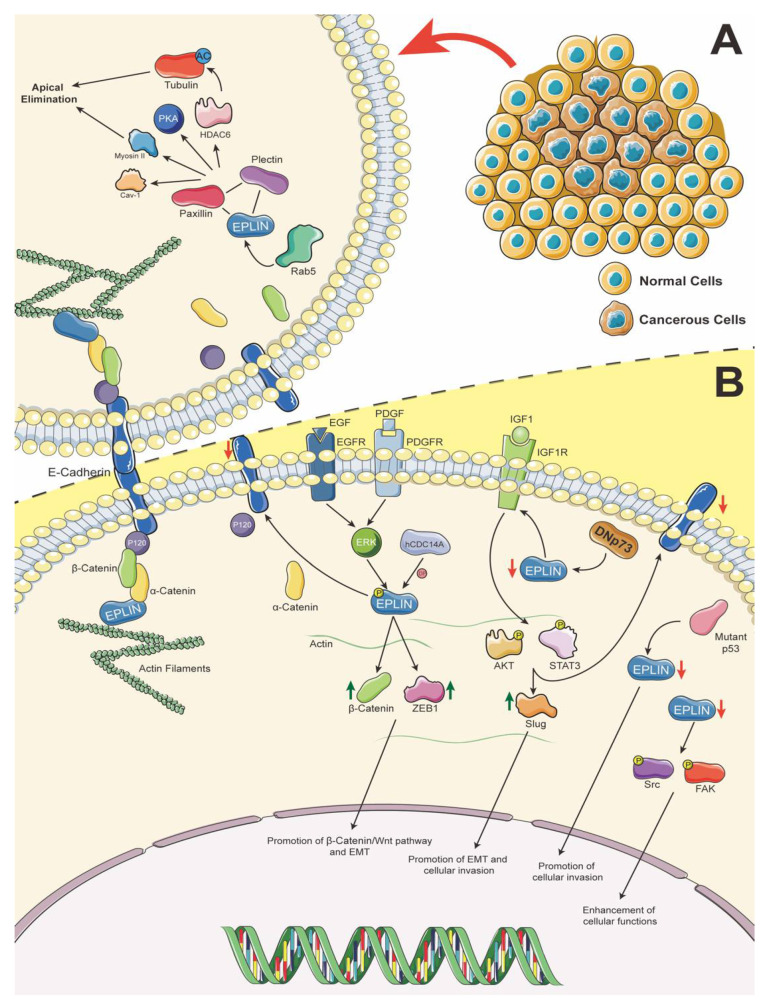
A. Role of EPLIN in apical extrusion. Rab5 allows disruption of the connection between the cadherin–catenin complex and EPLIN, allowing EPLIN to interact with paxillin and plectin to recruit/activate Cav-1, PKA and Myosin II, and acetylate tubulin through paxillin’s regulation of HDAC6 activity. Hence, promoting apical elimination when Ras-transformed cells are surrounded by normal cells. B. EPLIN prospective pathways in epithelium. EPLIN stabilises AJs and actin dynamics by binding to the cadherin–catenin complex and actin directly. PDGF/EGF could induce phosphorylation of EPLIN via ERK signalling pathways and results in disorganisation of AJs and interruption of actin dynamics, which further upregulates expression/translocation of β-catenin and ZEB1, diminishes expression of E-cadherin leading to activation of the β-catenin/Wnt pathway and promotion of EMT, further impacting cellular functions. This phosphorylation of EPLIN can be counteracted by hCDC14A. DNp73 induces downregulation of EPLIN, which allows IGF1R to interact with its ligand, then phosphorylate AKT and STAT3, which increases expression of Slug and decreases expression of E-cadherin, to contribute to promotion of the EMT process. p53 mutation can lead to downregulation of EPLIN expression, which results in enhancement of cellular invasiveness. The downregulation of EPLIN has been reported to promote cellular functions, which may be attributed to phosphorylation of FAK/Src and activation of FAK/Src pathways. Icons were obtained from SMART-Servier Medical ART (https://smart.servier.com, accessed on 18 April 2021) and graphics of pathways were designed and created using Adobe Illustrator 2021 Version 25.2.1 (Adobe Inc., San Jose, CA, USA).

**Figure 3 biomolecules-11-01038-f003:**
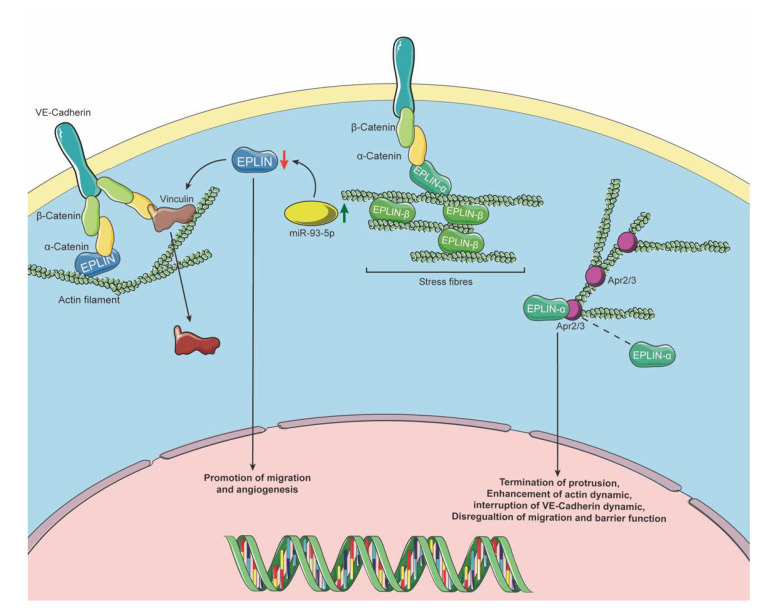
EPLIN’s role in endothelium. EPLIN sustains cell junctions in ECs by linking to VE-cadherin via α-catenin. Downregulation of EPLIN could be induced by upregulation of miR-93-5p, which may subsequently lead to translocation of vinculin away from cell junctions. Downregulation of EPLIN also results in promotion of cellular migration and supports angiogenesis. EPLIN-α has been reported to locate at JAIL and cLP and to regulate protrusion progression via the Arp2/3 complex, whilst EPLIN-β mainly locates at stress fibres to maintain its stability. Icons were obtained from SMART-Servier Medical ART (https://smart.servier.com, accessed on 18 April 2021) and graphics of pathways were designed and created using Adobe Illustrator 2021 Version 25.2.1 (Adobe Inc., San Jose, CA, USA).

**Table 1 biomolecules-11-01038-t001:** Novel Interacting and Regulatory Partners of EPLIN. For previously described interacting partners, please refer to review papers from Collins et al. [2] and Wu et al. [3].

Interacting/Regulatory Partners	Bio-Significances	Ref.
**p53**	p53 is a positive upstream regulator of EPLIN at the transcript level in osteosarcoma and lung cancer cells and at the protein level in lung, colon and breast cancer cells. p53 family transactivates the EPLIN gene on LIMA1-RE1 (AGGCAAGTTa tAACTgGCaT) and LIMA1-RE2 (GGACAgaaCT AGA-CAAGCCC). Depletion of EPLIN counteracts the repressed cancer invasion induced by p53 in lung cancer cells.	[16]
**hCDC14A**	Responsible for dephosphorylating EPLIN on Ser362 and Ser604. Knocking down EPLIN or reducing the activity of hCDC14A in HCT-16 cell lines leads to promotion of the EMT process.	[18]
**Cav-1**	Knocking down EPLIN expression will inhibit the accumulation/activity of these proteins in Ras12-transformed cells when surrounded by normal cells.	[20,23]
**Myosin II**		
**PKA**		
**Rab5**	Mutation of Rab5 could reduce accumulation of EPLIN in Ras-transformed cells surrounded by normal cells, while Rab5 acts upstream of EPLIN, mediates endocytosis of E-cadherin and allows EPLIN disconnection with the cadherin–catenin complex, leading to promotion of apical extrusion.	[22]
**Plectin**	Plectin and paxillin colocalise with EPLIN in Ras-transformed cells and have a mutual positive correlation and regulate apical extrusion.	[23,24]
**Paxillin**		
**FAK/Src**	EPLIN regulates invasion and migration in prostate cancer cells through the FAK/Src pathways.	[25]
**miR-93-5p**	miR-93-5p regulates EPLIN expression negatively by binding to its 3′-UTR sequence and associates with migration and angiogenesis in HUVECs.	[29]
**miR-221**	Depletion of miR-221 in PC3 cells results in enhancement of EPLIN protein level in combination with reduction of cellular migration and invasion.	[30]
**LUZP1**	LUZP1 interacts and colocalises with both isoforms of EPLIN in RPE-1 cells and regulates the ciliation process.	[32]
**NPC1L1**	EPLIN colocalises with NPC1L1 and myosin Vb on the peripheral brush border region of mouse small intestine. EPLIN interacts with NPC1L1 by binding its C164LG residues to the Q1277KR residues of NPC1L1. EPLIN also interacts with myosin Vb due to the interaction between the aa 21 to 40 regions of myosin Vb and the aa 491 to 511 regions of EPLIN. Mutant/depleted EPLIN has a positive effect on cholesterol absorption by disrupting the transportation ability of NPC1L1 in vivo.	[36]

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
