# Peer review of "Epithelial Protein Lost in Neoplasm, EPLIN, the Cellular and Molecular Prospects in Cancers"

_biomolecules, 2021, doi:10.3390/biom11071038_

Round 1

Reviewer 1 Report

In this review article, the authors summarize and discuss the recent progress in understanding biological functions of EPLIN protein and implications in cancer. Since EPLIN gene was discovered in 1998, the pubmed database has collected 54 references including 5 reviews. However, the latest review was published in 2017. Therefore,  this review article should be timely and informative to researchers.

It would be informative if the authors provide an additional figure describing a schematic structure of EPLIN protein (isoforms). If possible, include the location of the binding partners listed in Table-1 and post-translationally modified site(s).

Line 114-115: “The ability mouse EPLIN has to sustain actin dynamics…” ?

Line 148-149: “The activity of filamin in normal cells surrounding transformed cells is also repressed in this setting.” What kind of activity? Reference?

Resolution of the figures should be improved.

Author Response

Comment:

It would be informative if the authors provide an additional figure describing a schematic structure of EPLIN protein (isoforms). If possible, include the location of the binding partners listed in Table-1 and post-translationally modified site(s).

Author response:

We are grateful to the reviewer for this comment. Originally this was not included as it has previously been documented in past reviews and publications. However, we agree that the inclusion of this schematic enhances the review article. A schematic of EPLIN isoforms have been included as a new Figure 1, subsequent figure numbers have been changed accordingly.

We have also updated Table 1 to include additional information in regard to binding partners and residues.

Comment:

Line 114-115: “The ability mouse EPLIN has to sustain actin dynamics…” ?

Author response:

We have revised this sentence and section of the manuscript, it now reads:

The ability mouse EPLIN has to bind and regulate actin dynamics is reported to be weakened following ERK phosphorylation of serine residues within EPLIN, induced by platelet-derived growth factor (PDGF) [8]. A later study subsequently demonstrated the capacity of EGF induced ERK to phosphorylate human EPLIN in prostate cancer cells on Ser362 and Ser604, which pair to Ser360 and Ser602 in mouse EPLIN, leading to EPLIN ubiquitination and degradation and downregulation of E-cadherin, an essential marker of EMT progression [11].

Comment:

Line 148-149: “The activity of filamin in normal cells surrounding transformed cells is also

repressed in this setting.” What kind of activity? Reference?

Author response:
We are grateful to the reviewer for this comment. This section of the manuscript has been expanded and extended in accordance with the second reviewer’s comment for increase clarity.

The section previously reading:

“In a cell setting when Ras-transformed cells are surrounded with normal cells, EPLIN is discovered to be located predominantly at the peripheral region and cytoplasm of transformed cells and at the interphase between transformed cells and normal cells. Caveolin-1 (Cav-1), a positive player of the process, has been observed to be partly co-localised with EPLIN using immunoprecipitation assays and western blotting confirms a linkage between these two elements. Knocking down EPLIN by short hairpin RNA in transformed cells, the expression of Cav-1, myosin II and protein kinase A (PKA) in transformed cells were inhibited as well as the apical extrusion process. The activity of filamin in normal cells surrounding transformed cells is also repressed in this setting. Furthermore, using antagonists to inhibit MAPK/ERK Kinase (MEK) or actin polymerization leads to punctate accumulation of EPLIN and Cav-1 [19].”

Has been expanded to

“EPLIN has been indicated in the process of cytokinesis, in which loss of EPLIN and its interacting partners, could lead to multinucleation which in turn may contribute to car-cinogenesis [12, 13]. During the early carcinogenesis period in epithelial tissue, mutation of oncogenes are crucial contributors. Kajita et al. revealed, by using mammalian epi-thelial cells, that when Ras-transformed cells are surrounded by normal cells, the latter will fight to lift transformed cells from the monolayer in order to eliminate them. This competitive self-defence process was described as epithelial defence against cancer (EDAC) [19]. EPLIN has also been identified to play a role in this process, contributing to apical extrusion of RasV12-transformed Madin-Darby Canine Kidney (MDCK) cells, where it has been linked to Caveolin-1 (Cav-1) [20]. In the study, Ohoka et al. utilised Cav-1 immunoprecipitation under a number of cell culture conditions, namely normal cell culture, RasV12 transformed cell culture and combinations of the two cultures. The study has identified EPLIN as a binding partner predominantly in the mixed culture setting and at apical and lateral membrane domains, where it was observed to be partially co-localised with Cav-1 using immunofluorescence. EPLIN was also seen in the cytosolic and RasV12 intracellular regions where Cav-1 was absent. Knocking down EPLIN by short hairpin RNA in transformed cells, suppressed accumulation of Cav-1 and activation of myosin II and protein kinase A (PKA). It also impacted the apical extrusion process. Interestingly, Cav-1 knockdown did not impact EPLIN accumulation or myosin II and PKA activity. It is interesting to note that addition of the MAPK/ERK Kinase (MEK) in-hibitor U10126 or the actin polymerisation inhibitor, cytochalasin D, could impact the accumulation, localisation or enrichment of Cav-1 and EPLIN. The authors also demon-strate that the accumulation of filamin in normal cells surrounding transformed cells is repressed following EPLIN or Cav-1 knockdown in RasV12 transformed cells and simi-larly, EPLIN and Cav-1 enrichment in transformed cells is suppressed following filamin A knockdown in surrounding normal cells [20].”

Comment:

Resolution of the figures should be improved.

Author comment:

We apologise for the low resolution of the figures. This was caused during the initial submission, by embedding the images into the word document which only kept low resolution. Following revisions and updating of the figures or generation of new figures, high resolution images will be included with the revised manuscript. This include embedding images as tif files and uploading the original graphs in the submission system.

Reviewer 2 Report

In this manuscript, Zeng et al. provide a pertinent review of the recent findings on the biology of EPLIN. This protein plays critical roles in multiple cellular processes and has an impact on cancer development and possibly other diseases. The review tries to be comprehensive and touches upon recently discovered EPLIN molecular partners and biological roles. One interesting point is the realization that the two EPLIN isoforms have different functions and are differentially expressed and localized. There are also growing evidences that EPLIN may have cell and tissue-specific roles. These issues could be developed further, and clearer, in the manuscript, as it would help the reader grasp the biology of EPLIN better. I would suggest making a table indicating information on cell line/tissue and the respective abundance of each EPLIN isoform, plus any information about their specific functions in that context.

 Despite the relevance of the proposed review, prior to publication the manuscript requires substantial revision (e.g. in terms of English) and editing. Indeed, the wording is often confusing, the sentences too long, punctuation is lacking, etc. In some cases, I did not understand what the authors meant or found the statements not very correct. Some examples are listed below.

Lines 56 and 57:

“Interestingly, EPLIN is associated to early cellular event…”. What is early cellular event?

Lines 57 and 58:

“EPLIN was detected to co-located at…”. English needs correction.

Lines 128-132 – Very long and confusing sentence.

Lines 144 and 145:

“Caveolin (Cav-1), a positive player of the process, has been observed to be partly co-localized with EPLIN using immunoprecipitation assays and western blotting…”. Immunoprecipitation and western blotting are not used to determine protein co-localization. Indeed if two proteins are shown to interact by immunoprecipitation chances are that they co-localize but this is assessed by imaging studies.

Lines 283 and 284:

“Goncalves et al. identified EPLIN as a potential interacting partner of LUZP1 by conducting affinity purification and mass spectrometry assays in a HeLa cells database.”. Written in a very confusing way.

These are just some examples and more can be found throughout the text. In sum, the manuscript needs extensive revisions to make sure the reviewed data was extracted properly from the original publications, and communicated in a clear manner.

Author Response

Reviewer 2:

Comment:

In this manuscript, Zeng et al. provide a pertinent review of the recent findings on the biology of EPLIN. This protein plays critical roles in multiple cellular processes and has an impact on cancer development and possibly other diseases. The review tries to be comprehensive and touches upon recently discovered EPLIN molecular partners and biological roles. One interesting point is the realization that the two EPLIN isoforms have different functions and are differentially expressed and localized. There are also growing evidences that EPLIN may have cell and tissue specific roles. These issues could be developed further, and clearer, in the manuscript, as it would help the reader grasp the biology of EPLIN better. I would suggest making a table indicating information on cell line/tissue and the respective abundance of each EPLIN isoform, plus any information about their specific functions in that context.

Author response:

We are grateful to the reviewer for their comments and suggestions. We agree that the implications of the isoform specific roles is a significant and important finding in this area of research. Whilst the main outline of the current review is focused on EPLIN in a cancer context, we agree that this is a key area for discussion. To address this we have greatly expanded the section reporting the important findings of Taha et al in relation to the isoform specific roles of EPLIN in endothelial cells, and indeed other sections that required clarification.

This section now reads:

“In their study, Taha et al. focused on the specific function of individual isoforms in endothelial cells. The authors reported expression of both α and β isoforms in pig aorta and cava vein endothelial cells noting enhanced expression of EPLINβ in aorta compared to cava vein sample, with little change noted in EPLINα between samples. Interestingly they report enhanced EPLINβ expression in confluent HUVECs following sheer stress application, whilst EPLINα levels were not significantly impacted. Additionally, EPLINα but not β isoform expression was noted to correlate with HUVEC confluence, with its expression, protrusion formation and migration velocity reducing in confluent compared to growing cultures. EPLIN ex-pression was observed to be localised to cell junction and stress fibres with analysis of tagged isoform localisation also indicating their presence at cell junctions and stress fibres. Importantly, the authors observed an isoform specific role for EPLINα, noting its presence within or in close proximity to branched actin filament networks at membrane protrusions, such as classical lamellipodia (cLP) and junc-tion-associated intermittent lamellipodia (JAIL). However, EPLINβ was mostly negative at protrusions and was noted to be potentially involved in retraction. EPLINα was also observed to be more dynamic than EPLINβ, collectively supporting differential isoform roles in endothelial cells and the proposal that EPLINα contributes to cell migration and junction remodelling whereas EPLINβ con-tributes to filament stabilisation [42].

Subsequently, Taha et al. investigated the role of EPLINα at cLP and JAIL and its relationship with the Arp2/3 complex. JAIL is responsible for forming and developing endothelial cells junction sites, where vascular endothelial cadherin (VE-cadherin) lo-calises. This dynamic regulation by JAIL contributes to migration and junction dynamics related to angiogenesis [43]. Spinning disc, confocal microscopy (SpDM) demonstrated that the distance between EPLIN-α and the Arp2/3 complex narrowed during protrusion extension and that the eventual overlap of these protein resulted in the halting of pro-trusion movement. This coincided with EPLINα and Arp2/3 disconnection with and loss of actin filaments and combined with the findings that EPLIN isoforms could associate with the Arp2/3 complex during pull down assays, implicated a potential role for EPLIN in protrusion termination via the Arp2/3 complex interaction. In support of this, the authors demonstrate that CK666 mediated inhibition of Arp2/3 resulted in an inhibition of pro-trusion formation and EPLINα’s materialisation at the membrane. In keeping with this, targeting EPLIN using siRNA resulted in both increased protrusion size and duration and an enhancement of migration velocity in normal medium. Furthermore, overexpressing of EPLIN-α-EGFP resulted in EPLIN-α-EGFP appearance at and disruption of JAIL like structures, preventing protrusion expansion, and also enhanced the formation of filopodia, indicating enhancement of actin dynamic. The authors suggest that enhanced EPLINα results in elevated binding to the Arp2/3 complex and JAIL formation termination, potentially impacting on VE-cadherin dynamics. In keeping with this, they further demonstrate altered VE-cadherin dynamics in EPLINα overexpression HUVEC cells, observing intracellular gaps and disrupted VE-cadherin at cell contacts, together with decreased migration and barrier function[42]. Interestingly, the authors demonstrate a prominent role of EPLIN-β in the stress fibre induction and stabilisation demonstrating stress fibre formation is enhanced substantially following EPLINβ overexpression (and to a lesser extent EPLINα overexpression) in endothelial cells. Furthermore, EPLIN-β was found to protect stress fibres from depolymerisation resulting from Y27632 ROCK inhibitor treatment, with minimal disassembly noted in EPLINβ overexpression cells compared to control or EPLINα overexpression cells following such treatment [42]. This important study has shed light on the isoform specific role of EPLIN in endothelial cells, highlighting EPLIN-α’s role in regulating protrusion progressions by interacting with Arp2/3 and regulating JAIL formation, VE-cadherin dynamics and implication in migration and barrier function, whereas EPLIN-β plays key roles in induction and stabilisation of stress fibres. Currently, our understanding of the isoform specific role of EPLIN is limited. Further investigations in this vital area are needed to enhancing our under-standing of this important molecule across many biological areas including cancer development and progression.

Taken together, EPLIN takes part in the regulation of endothelial dynamics by binding to VE-Cadherin via α-catenin and actin filaments [39]. Depletion of EPLIN could be induced by miR-93-5p which contributes to elevation of cellular migration and lumen formation [29] whereas overexpression of EPLINα reduced cell migration and tubule formation in HECV endothelial cells [41]. Hence, EPLIN appears to play key roles in endothelium cells and associated processes. A summary outlining the potential role of EPLIN in endothelial cells is outlined in Figure 3.”

Comment:

Despite the relevance of the proposed review, prior to publication the manuscript requires

substantial revision (e.g. in terms of English) and editing. Indeed, the wording is often confusing, the sentences too long, punctuation is lacking, etc. In some cases, I did not understand what the authors meant or found the statements not very correct.

Some examples are listed below.

Lines 56 and 57:

“Interestingly, EPLIN is associated to early cellular event…”. What is early cellular event?

Lines 57 and 58:

“EPLIN was detected to co-located at…”. English needs correction.

Lines 128-132 – Very long and confusing sentence.

Lines 144 and 145:

“Caveolin (Cav-1), a positive player of the process, has been observed to be partly co-localized with EPLIN using immunoprecipitation assays and western blotting…”. Immunoprecipitation and western blotting are not used to determine protein co-localization. Indeed if two proteins are shown to interact by immunoprecipitation chances are that they co-localize but this is assessed by imaging studies.

Lines 283 and 284:

“Goncalves et al. identified EPLIN as a potential interacting partner of LUZP1 by conducting affinity purification and mass spectrometry assays in a HeLa cells database.”.

Written in a very confusing way.

These are just some examples and more can be found throughout the text.

Author response:

We thank the reviewer for pointing out the issues on readability and clarity. We have undertaken substantial revision of the manuscript to try to address such concerns and check accuracy. We have noted and corrected several issues related to language, reference links and accuracy/clarity and have corrected these. Furthermore, we have had additional members of the team check and correct the manuscript for its grammar, punctuation and English (this has been acknowledged in the included acknowledgement sections in the revised manuscript).

In response to specific examples:

1) Interestingly, EPLIN is associated to early cellular event…”.

Has been clarified to

“Interestingly, EPLIN was found to be essential for cell division”

2) “EPLIN was detected to co-located at…”

Has been altered to

“EPLIN was detected to locate at the cleavage furrow and to associate with myosin II and Sept2 during ingression period in Hela and MCF-7 cells”

3) The long sentence (128-132) and sentence before has been incorporated together to provide a clearer narrative. Collectively it now reads:    

“Knocking down EPLIN and hCDC14A in HCT-116 CRC cell lines decreased E-cadherin significantly to acquire mesenchymal characteristics [18]. The study also demonstrated that downregulation of EPLIN and hCDC14A were associated with poor prognosis in colorectal cancer, by exploring online databases. Thus, it would appear that hCDC14A acts as a vital upstream player of EPLIN during cancerous development.”

4) “Caveolin (Cav-1), a positive player of the process, has been observed to be partly co-localized with EPLIN using immunoprecipitation assays and western blotting”

We are grateful to the reviewer for their comments regarding the technical assessment of localisation and need for microscopy studies associated with this sentence. We have focused on this section to expand the study and method descriptions and provide clarity:

The section:

“In a cell setting when Ras-transformed cells are surrounded with normal cells, EPLIN is discovered to be located predominantly at the peripheral region and cytoplasm of transformed cells and at the interphase between transformed cells and normal cells. Caveolin-1 (Cav-1), a positive player of the process, has been observed to be partly co-localised with EPLIN using immunoprecipitation assays and western blotting confirms a linkage between these two elements. Knocking down EPLIN by short hairpin RNA in transformed cells, the expression of Cav-1, myosin II and protein kinase A (PKA) in transformed cells were inhibited as well as the apical extrusion process. The activity of filamin in normal cells surrounding transformed cells is also repressed in this setting. Furthermore, using antagonists to inhibit MAPK/ERK Kinase (MEK) or actin polymerization leads to punctate accumulation of EPLIN and Cav-1 [19]. Hence, EPLIN is implicated to take part in this interesting competitive process along with a number of elements and Mitogen activated Protein Kinase (MAPK) pathways and actin dynamic are indicated to be involved”

Has been changed to:

“EPLIN has also been identified to play a role in this process, contributing to apical extrusion of RasV12-transformed Madin-Darby Canine Kidney (MDCK) cells, where it has been linked to Caveolin-1 (Cav-1) [20]. In the study, Ohoka et al. utilised Cav-1 immunoprecipitation under a number of cell culture conditions, namely normal cell culture, RasV12 transformed cell culture and combinations of the two cultures. The study has identified EPLIN as a binding partner predominantly in the mixed culture setting and at apical and lateral membrane domains, where it was observed to be partially co-localised with Cav-1 using immunofluorescence. EPLIN was also seen in the cytosolic and RasV12 intracellular regions where Cav-1 was absent. Knocking down EPLIN by short hairpin RNA in transformed cells, suppressed accumulation of Cav-1 and activation of myosin II and protein kinase A (PKA). It also impacted the apical extrusion process. Interestingly, Cav-1 knockdown did not impact EPLIN accumulation or myosin II and PKA activity. It is interesting to note that addition of the MAPK/ERK Kinase (MEK) inhibitor U10126 or the actin polymerisation inhibitor, cytochalasin D, could impact the accumulation, localisation or enrichment of Cav-1 and EPLIN. The authors also demonstrate that the accumulation of filamin in normal cells surrounding transformed cells is repressed following EPLIN or Cav-1 knockdown in RasV12 transformed cells and similarly, EPLIN and Cav-1 enrichment in transformed cells is suppressed following filamin A knockdown in sur-rounding normal cells [20]. Hence, EPLIN has been implicated to take part in this interesting competitive process along with a number of elements and Mitogen Activated Protein Kinase (MAPK) pathways and actin dynamic are indicated to be involved.”

“Goncalves et al. identified EPLIN as a potential interacting partner of LUZP1 by conducting affinity purification and mass spectrometry assays in a HeLa cells database.”.

Has been altered to:

“Goncalves et al. identified EPLIN as a potential interacting partner of LUZP1 by conducting BioID assays in cycling and serum- deprived HEK293 cells”

Comment:

In sum, the manuscript needs extensive revisions to make sure the reviewed data was extracted properly from the original publications, and communicated in a clear manner

Author response:

In revising the manuscript we have addressed the examples listed by the reviewer. In addition to this we have undertaken additional checks on the manuscript for accuracy and clarity. A number of sections have been expanded and we have corrected a number of noticed errors or reference mis-links. We are very grateful to the reviewer for highlighting such issues and hope that our substantial manuscript revisions have addressed all reviewer concerns in regard to these comments.

Round 2

Reviewer 2 Report

The authors have significantly improved the manuscript and have addressed most of the concerns raised during the first revision. However, there are still some minor issues:

1 – p53 is indicated as an interacting partner. This usually indicates that two proteins interact. However, what has been reported is that p53 is a transcriptional regulator of EPLIN expression, which is something different. In a quick search in BioGrid I found that p53 and EPLIN were shown to interact in affinity capture-mass spectrometry, but to my knowledge these interactions were not validated nor their biological significance studied. I recommend that the authors either do not refer to p53 as an interacting partner or mention the possibility that it interacts with EPLIN but this still needs further investigation.

2 – The manuscript still needs a meticulous revision in terms of English to address some remaining issues.

Examples:

Line 303 – I believe the authors mean GFP-tagged instead of targeted

Line 307 – Replace eligible with able or synonym.

Line 321 – Consider revising the expression biological progressions.

Line 395 – A GST pull-down assay and not assays.

Line 400 – “which is promotes” should be “which promotes”.

Lines 438-441 – Sentence is too long and confusing.

Line 448 – Actin dynamics and not dynamic.

Line 450 – Consider replacing the expression “In keeping with this” as it was used two sentences prior.

Line 484 – “Our lab, as a contributors to” should be “as a contributor to”.

In conclusion, if the authors address these minor issues, I recommend that the manuscript be accepted for publication.

Author Response

Response to Reviewer-2

Reviewer Comment:

p53 is indicated as an interacting partner. This usually indicates that two proteins interact. However, what has been reported is that p53 is a transcriptional regulator of EPLIN expression, which is something different. In a quick search in BioGrid I found that p53 and EPLIN were shown to interact in affinity capture-mass spectrometry, but to my knowledge these interactions were not validated nor their biological significance studied. I recommend that the authors either do not refer to p53 as an interacting partner or mention the possibility that it interacts with EPLIN but this still needs further investigation.

Author response:

We are very grateful to the reviewer for raising this important issue and agree that the language used needs to be altered for clarity.

We have made a number of revisions to the p53 section in line with this comment. These are as follows:

Line 91 – 93 

DNp73, a mutant isoform of the p53 family capable of inhibiting the expression of p73, was found to be involved in the EPLIN interacting network

Has been altered to:

DNp73, a mutant isoform of the p53 family capable of inhibiting the expression of p73, was found to be involved in the regulation of EPLIN

Line 99 – 102

Due to the relationship between DNp73 and p53, it naturally draws the attention to whether p53 also takes part in this interacting network of EPLIN. Indeed, later study by Ohashi et al. [16] uncovered the association between these two tumour suppressors

Has been amended to:

Due to the relationship between DNp73 and p53, it naturally draws the attention to whether p53 also takes part in the regulation of EPLIN. Indeed, later study by Ohashi et al. [16] uncovered the regulatory association between these two tumour suppressors

Line 120 – 121

Hence, EPLIN was reported to be a target of p53, with such interactions influencing metastatic progression.

Has been amended to:

Hence, EPLIN was reported to be a target of p53, with this relationship influencing metastatic progression.

Figure legend 1, the sentence:

“p53 mutation down-regulates expression of EPLIN”

Has been altered to:

“p53 mutation can lead to down-regulation of EPLIN expression”

Additionally, we have modified Table 1. This was originally listed as novel interacting partners of EPLIN. However, in line with the reviewer’s comments, we agree that the terminology regarding interacting partners likely implies direct interactions between the two proteins. Looking again at the table, following the comments related to p53, (which is more thoroughly implicated in a regulatory role, regulating EPLIN expression) we agree better terminology should be used for clarity. This would also be more suitable in terms of the relationship between Rab5, Src, FAK, miR-93-5-p, Mir221 etc where largely the predominant relationship is through expression regulation or little evidence is available to demonstrate a direct protein-protein interaction with EPLIN.

Therefore, we have altered the table’s focus to indicate “Interacting or regulatory partners of EPLIN”

In relation to this, the table legend has changed from:

“Table 1. Novel Interacting Partners of EPLIN. For previously described interacting partners, please refer to review papers from Collins et al. [2] and Wu et al. [3].”

To:

“Table 1. Novel Interacting and Regulatory Partners of EPLIN. For previously described interacting partners, please refer to review papers from Collins et al. [2] and Wu et al. [3].”

The linker sentence to Table 1 in the text (line 357-358) has been altered from:

“Taken together, the above studies have increased our understanding of the interacting networks associated with EPLIN, and these are outlined in Table 1.”

To:

“Taken together, the above studies have increased our understanding of the interacting and regulatory networks associated with EPLIN, which are outlined in Table 1.”

The column heading “Interacting Partners” in Table 1 has been altered to “Interacting/Regulatory Partners”

Finally, several locations in the manuscript have been altered for clarity relating to the EPLIN relationship (changes highlighted in bold).

Line 73: Now reads:

“2. New members of EPLIN’s interacting/regulatory network”

Line 84-86 now reads:

“To understand more about its role in these processes, and others, increasing research has focused on elucidating interacting or regulatory proteins of EPLIN, either upstream regulators or downstream participants.”

Line 404 – 406 now reads:

This may imply that the role of EPLIN in HUVECs cellular functions could be achieved through interaction with, or regulation by, other elements.

Line 512-514 now reads:

Furthermore, research has identified a number of interacting partners or regulatory mechanisms associated with EPLIN

Reviewer Comment:

2 – The manuscript still needs a meticulous revision in terms of English to address some remaining issues.

Examples:

Line 303 – I believe the authors mean GFP-tagged instead of targeted

Line 307 – Replace eligible with able or synonym.

Line 321 – Consider revising the expression biological progressions.

Line 395 – A GST pull-down assay and not assays.

Line 400 – “which is promotes” should be “which promotes”.

Lines 438-441 – Sentence is too long and confusing.

Line 448 – Actin dynamics and not dynamic.

Line 450 – Consider replacing the expression “In keeping with this” as it was used two sentences prior.

Line 484 – “Our lab, as a contributors to” should be “as a contributor to”.

Author Response:

We are again grateful to the reviewer for highlighting these English related issues and have undertaken additional review to further improve throughout (tracked in revised manuscript). The manuscript has been subjected to additional checks by the authors and also other members of the team (Dr Jane Lane and Ms Fiona Ruge – updated in the acknowledgements section). We have also corrected those changes outlined in the reviewer examples as listed below:

Line 303 – I believe the authors mean GFP-tagged instead of targeted

Amended to “GFP-tagged”

Line 307 – Replace eligible with able or synonym.

“eligible” has been changed to “able”

Line 321 – Consider revising the expression biological progressions.

“biological progressions” has been revised to “biological processes”

Line 395 – A GST pull-down assay and not assays.

“assays” corrected to “assay”

Line 400 – “which is promotes” should be “which promotes”.

Corrected to “which promotes”

Lines 438-441 – Sentence is too long and confusing.

Spinning disc, confocal microscopy (SpDM) demonstrated that the distance between EPLIN-α and the Arp2/3 complex narrowed during protrusion extension and that the eventual overlap of these protein resulted in the halting of protrusion movement. This coincided with EPLINα and Arp2/3 disconnection with and loss of actin filaments and combined with the findings that EPLIN isoforms could associate with the Arp2/3 complex during pull down assays, implicated a potential role for EPLIN in protrusion termination via the Arp2/3 complex interaction.

Has been revised to include additional punctuation to

Spinning disc, confocal microscopy (SpDM) demonstrated that the distance between EPLIN-α and the Arp2/3 complex narrowed during protrusion extension, and that the eventual overlap of these proteins resulted in the halting of protrusion movement. This coincided with EPLINα and Arp2/3 disconnection with, and loss of, actin filaments. Taken together with the findings that EPLIN isoforms associated with the Arp2/3 complex during pull down assays, implicated a potential role for EPLIN in protrusion termination via the Arp2/3 complex interaction.

Line 448 – Actin dynamics and not dynamic.

Corrected to “dynamics”

Line 450 – Consider replacing the expression “In keeping with this” as it was used two sentences prior.

In keeping with this has been altered to “Consistent with this”

Line 484 – “Our lab, as a contributors to” should be “as a contributor to”.

Amended to “contributor”

Reviewer comment:

In conclusion, if the authors address these minor issues, I recommend that the manuscript be accepted for publication.

Author Response:

We would again like to express our gratitude to the reviewer for undertaking a thorough review of our article and for their time in reviewing the revised version. We are very grateful for their comments related to our earlier version and that the concerns raised had been addressed, with only minor aspects remaining. We are very happy to revise the manuscript in accordance with these comments and hope that these revisions address the remaining issues. We feel that the input of the reviewer and their important comments (in both review rounds) have greatly improved the review article.